# Structures of the CcmABCD heme release complex at multiple states

Jiao Li [1,2,4], Wan Zheng[1,4], Ming Gu [1], Long Han[2], Yanmei Luo[1], Koukou Yu[1], Mengxin Sun[1], Yuliang Zong[1], Xiuxiu Ma[1], Bing Liu[1], Ethan P. Lowder [3], Deanna L. Mendez[3], Robert G. Kranz [3] ✉, Kai Zhang [2] ✉ & Jiapeng Zhu [1] ✉

Cytochromes *c* use heme as a cofactor to carry electrons in respiration and photosynthesis. The cytochrome *c* maturation system I, consisting of eight membrane proteins (CcmABCDEFGH), results in the attachment of heme to cysteine residues of cytochrome *c* proteins. Since all *c*-type cytochromes are periplasmic, heme is first transported to a periplasmic heme chaperone, CcmE. A large membrane complex, CcmABCD has been proposed to carry out this transport and linkage to CcmE, yet the structural basis and mechanisms underlying the process are unknown. We describe high resolution cryo-EM structures of CcmABCD in an unbound form, in complex with inhibitor AMP-PNP, and in complex with ATP and heme. We locate the ATP-binding site in CcmA and the heme-binding site in CcmC. Based on our structures combined with functional studies, we propose a hypothetic model of heme trafficking, heme transfer to CcmE, and ATP-dependent release of holoCcmE from CcmABCD. CcmABCD represents an ABC transporter complex using the energy of ATP hydrolysis for the transfer of heme from one binding partner (CcmC) to another (CcmE).

Cytochromes *c* play a central role in electron transport in respiration and photosynthesis[1–6]. In addition, they are found to play other important functions such as activation of programmed cell death in eukaryotes (apoptosis)[7,8]. Cytochromes *c* differ from other cytochromes by at least one thioether bond of the heme vinyl group with the sulfur atom of the cysteine residue in the protein. Typically, there are two thioether bonds between two vinyl groups in heme and two cysteinyls in a CXXCH "heme-binding motif" in the protein, with the histidine in the same motif forming an axial ligand to the heme iron atom[9,10]. The covalent attachment leads to very stable heme proteins with a wide range of redox potentials[11]. Indeed, the CXXCH attachment motif has been used in engineering novel cytochrome *c* catalysts, as long as a pathway for heme attachment is overexpressed with these cytochromes *c*[12–14].

All cytochromes *c* are located outside the inner membrane, therefore the maturation of cytochromes *c* requires the transport of

apocytochrome polypeptide and heme across the inner membrane and a membrane-bound machinery that attaches heme to the sulfur atom(s) of the cysteine residue(s). All cytochromes *c* in bacteria have a SEC-dependent signal sequence for export, where periplasmic thiol reduction proteins maintain the thiols in the reduced form, a prerequisite for attachment by the cytochrome *c* synthases.

Cytochrome *c* maturation can be catalyzed by three pathways: system I (CcmABCDEFGH), system II (CcsBA), and system III (HCCS)[15–17]. Cytochrome *c* maturation (Ccm)-System I catalyzes the most complex process, found in α- and γ-proteobacteria, archaea, and plant and some protozoal mitochondria[18]. System I from *Escherichia coli* is comprised of eight membrane bound components (CcmABC-DEFGH), with the periplasmic thiol reductase CcmG involved in maintaining the reduced thiols of CXXCH. The remaining Ccm proteins are proposed to mediate two general steps (Supplementary Fig. 1 on complete pathway showing the two steps): step 1 is the CcmABCD-

[1]School of Medicine & Holistic Integrative Medicine, Nanjing University of Chinese Medicine, Nanjing 210023, China. [2]Department of Molecular Biophysics and Biochemistry, Yale University, New Haven, CT 06511, USA. [3]Department of Biology, Washington University in St. Louis, CB 1137, One Brookings Drive, St. Louis, MO 63130-4899, USA. [4]These authors contributed equally: Jiao Li, Wan Zheng. ✉e-mail: kranz@wustl.edu; jack.zhang@yale.edu; zhujiapeng@hotmail.com

based formation of a holo(heme) CcmE, a periplasmic heme chaperone[19,20]. Step two is mediated by a cytochrome *c* synthase complex (CcmF/H)[19] that removes the heme from CcmE and attaches it to cyt *c* (CXXCH). Step one involves heme trafficking to an external heme binding domain in CcmC, whereby the heme is covalently attached to CcmE, and subsequently released to move freely in the periplasm. This release is mediated by a CcmABCD complex, which has been referred to as an ABC transporter release complex[21]. CcmA is a classic ATP-binding cassette protein that complexes with its integral membrane partners CcmB and CcmC.

The structural basis for step 1 is unknown, even the stoichiometry of the CcmABCD complex is unclear. Here we determine Cryo-EM structures of CcmABCD in the presence and absence of heme and ATP substrates to unravel the structural basis for how this unique ABC transporter release complex functions. The structures of CcmABCD intermediates in the heme trafficking and release cycle explain the heme release mechanisms, with genetic analyses that support the mechanisms. Conformational changes mediated by ATP hydrolysis of CcmA show how CcmB and CcmC movements would weaken the heme binding in CcmCD, releasing heme (holoCcmE) and preparing for

another cycle. Thus, with the recently published structure of CcmF[22], a component of the CcmF/H cyt *c* synthase, our CcmABCD structures elucidate the structural basis for this complicated system, a pathway present in all domains of life.

## Results

### Biochemistry and overall structure of CcmABCD

A hexahistidine-tagged CcmA was used to isolate the CcmABCD complex. IMAC (immobilized metal-affinity chromatography) and FPLC SEC (fast protein liquid chromatography size exclusion chromatography) were used to purify the complex, with Cryo-EM analysis on the "as purified" CcmABCD showing a composition of $CcmA_2B_2C_1D_1$ (Fig. 1a). We summarize here the conformations determined in the absence and presence of substrates (heme and ATP). The cytoplasmic CcmA is an ATP-binding cassette (ABC) protein. In the ATP binding site, a $Mg^{2+}$ is coordinated with six atoms: two oxygen atoms from the ATP, $N(\delta)$ of $H81^A$, $O(\beta)$ of $T41^A$ and two waters (Supplementary Fig. 2b, c). Other residues, $R14^A$, $R11^A$, $K40^A$, $T42^A$, $G37^A$ and $H184^A$ are in close contact with the ATP, which may play important role in stabilizing or hydrolyzing the ATP (Fig. 1b and Supplementary Fig. 2). CcmB is an

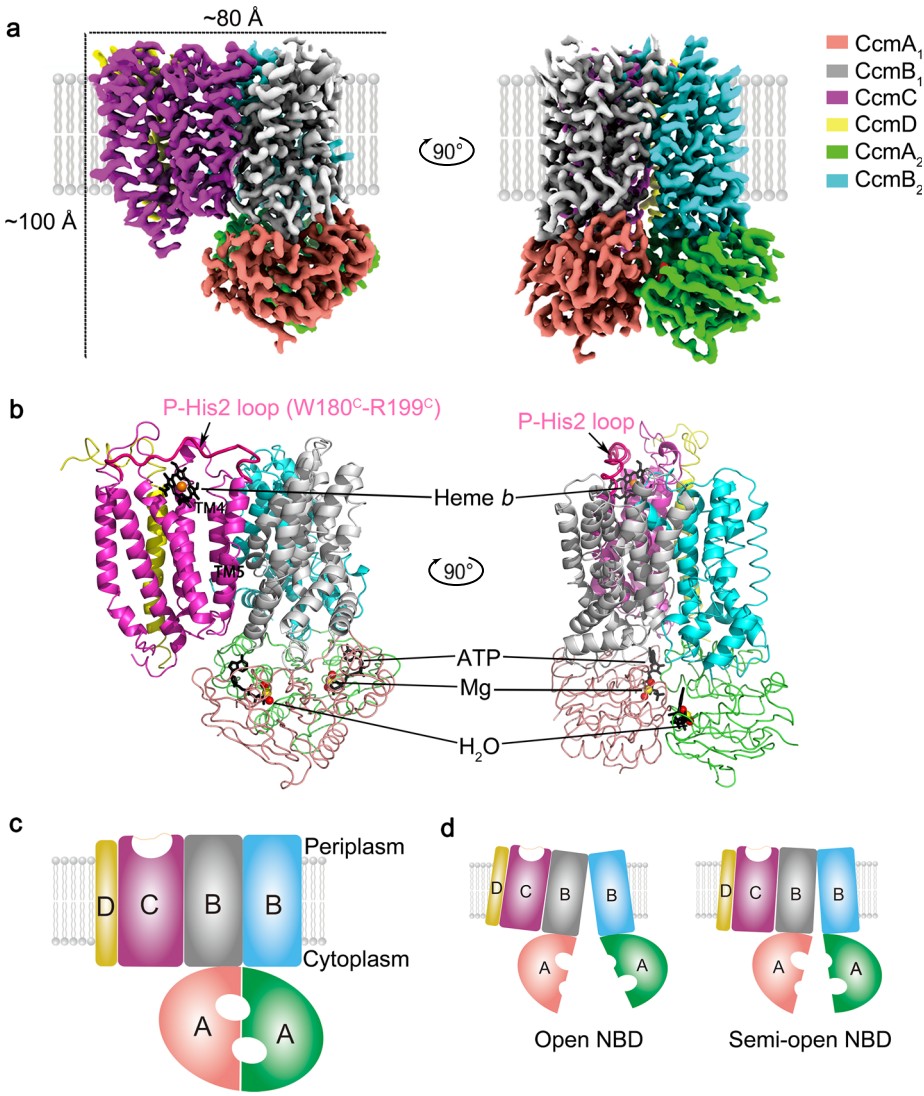

**Fig. 1 | Overall structure of the *E. coli* cytochrome *c* maturation complex CcmABCD. a** The cryo-EM density map of CcmABCD complex bound with heme *b* and two ATPs in closed NBD state. **b** Cartoon representation of CcmABCD structure. The bound ATPs and heme *b* are shown in black stick. The iron atom is shown in brown sphere. The highly flexible P-His2 loop of W180-R199 residues in CcmC is colored in red. **c** Schematic representation of closed NBD conformation. **d** Schematic representation of open NBD and semi-open NBD conformation. All of the three conformations are observed in our cryo-EM structures.

integral membrane protein containing six transmembrane (TM) helices, with CcmA and CcmB forming a homodimer, which is analogous to ABC transporters. CcmC interacts with CcmB subunits via two transmembrane helices CcmC-TM4 and TM5 (Fig. 1b). The small single TM protein CcmD[23] is bound to CcmC only. CcmC contains six TM helices and harbors a heme-binding site located at the external periplasmic interface (Fig. 1b). The cryo-EM structures of CcmABCD in complex with ATP and heme clearly reveal a ferrous heme in the binding site. A highly flexible loop of residues W180-R199 in CcmC (Fig. 1b), called the P-His2 loop, shows very low density in the Cryo-EM map, and thus may undergo conformational change during heme-binding and release (see below). CcmE, a heme chaperone protein which shuttles heme from CcmC to the cytochrome $c$ protein, was co-expressed with CcmABCD, but not observed in the SDS-PAGE gel (Supplementary Fig. 3a) or the cryo-EM structures (Fig. 1a). Our structures comprise three different conformations we classify based on open, semi-open or closed NBDs (nucleotide binding domains, Fig. 1c, d). Next, we evaluate the locations and dynamics of heme in CcmABCD.

## Heme-bound structures and heme trafficking

Previous studies have shown that when CcmA and/or B are deleted (or CcmA inactive), a stable CcmCDE complex with endogenous heme is isolated[24–26]. Mutational and spectroscopic analyses have demonstrated that CcmCDE possesses heme in the CcmC external (periplasmic) heme binding site, with heme liganded to His60$^C$ (called P-His1) and His184$^C$ (called P-His2)[25]. Heme in the complex is attached covalently to His130 of CcmE, just as in the isolated, released holoCcmE. These data provide support for the proposal that CcmA and CcmB are required for release of holoCcmE from the CcmCDE complex[21].

To begin to understand the structural basis for heme trafficking, binding, and holoCcmE release, we added heme and ATP to the CcmABCD preparations used for cryoEM and collected images for analyses. Two structures with this exogenous heme were resolved (Fig. 2a), showing that P-His1 (His60$^C$) is indeed an axial ligand to the heme, and that a conserved tryptophan-rich region, termed the WWD domain (Fig. 2c)[19], interacts with the edge of heme. Heme in the closed NBD state has P-His1 liganded to the heme iron at a distance of 2.8 Å and the bond between N($\delta$) of P-His1 and the heme iron is perpendicular to the heme ring (Fig. 2b). In a second conformation with heme, in a semi-open NBD state, P-His1 is not an axial ligand, for the distance between heme iron and P-His1 is 4.4 Å and the angle between the line of P-His1 N($\delta$)–heme iron and the heme plane is ~50° (Fig. 2b). In both conformations with heme, P-His2 is not a ligand and the entire P-His2 loop is located near the CcmC:B interface (Fig. 2d). We propose that we have trapped heme in conformations whereby the CcmABCD cycle of ATP hydrolysis weakens interactions with heme leading towards release. This includes weakening of the P-His ligands for switching to the holoCcmE ligand (i.e., Tyr134$^E$), and weakening of heme binding by the WWD domain.

Although a CcmABCD structure with holoCcmE was not isolated, we were able to use an existing cryo-EM structure of CcsBA[27], one of the three WWD family members (CcsBA, CcmC, CcmF, Fig. 2c), to reveal CcmABCD with heme liganded by P-His1 and P-His2. Each of the three WWD family members also have conserved P-His1 and P-His2, shown spectrally in CcmC to ligand heme in a CcmCDE complex[25]. The cryo-EM structure CcsBA was shown to have two conformations, open and closed, with "open" possessing heme in the external binding site[27]. Using the CcsBA open state, it is observed that heme is nearly identical in location with respect to the WWD domain and P-His1 in CcmC (Supplementary Fig. 4a compared to 4d). The CcsBA open state thus provides a framework for revealing the CcmC external binding site structure when both P-His ligands are coordinated to heme (Fig. 2e). Genetic evidence[25] has indicated that, like CcsBA[27], CcmC requires both P-His ligands for heme binding and in the case of CcmC for

holoCcmE formation. A comparison of CcmABCD structures with heme liganded by P-His1 (Fig. 2d) or both P-His residues (Fig. 2e) to non-heme bound states suggests that the P-His2 loop, as well as CcmC TM5, may undergo a major conformational change. Conformational changes between the various cryo-EM structures of CcmABCD show clear movements in TM5 (see below). We hypothesize that CcmC TM5 and P-His2 loop interactions with CcmB represent the structural basis for holoCcmE release during the ATPase cycle.

In order to better understand the roles of individual residues in CcmC, particularly heme trafficking, we tested selected CcmC variants for two functional activities (Table 1 and Supplementary Fig. 5). Cyt $c$ synthesis function implies that holoCcmE is formed and released (by CcmABCD) to the CcmF/H cyt $c$ synthase. HoloCcmE formation determines whether heme is covalently attached to His130 of CcmE. If no cyt $c$ is synthesized but holoCcmE is formed, the implication is that there is a holoCcmE release defect. We first tested possible histidine ligands in CcmC. Four histidine residues H60$^C$, H147$^C$, H174$^C$ and H184$^C$ on CcmC were each mutated to Ala. While H147A$^C$ and H174A$^C$ showed both activities, H60A$^C$ (P-His1) and H184A$^C$ (P-His2) could neither produce holoCcmE nor holo-cyt $c$ (Table 1 and Supplementary Fig. 5). These results confirm previous results[25], and we conclude that both are likely heme ligands, required for holoCcmE formation. We further interrogated the P-His2 loop region. G186$^C$ and S187$^C$ are the first two residues of the flexible loop on CcmC. Mutants that change either of these residues to alanine showed wild type phenotype. Deletion of either of G186$^C$ or S187$^C$ that shortens the flexible loop significantly reduced the activities: compared with the wild type, ΔG186$^C$ produced the same amount of holoCcmE, but only ~56% of holo-cyt $c$; whereas ΔS187$^C$ mutant totally abolished holo-cyt $c$ production and reduced holoCcmE. Results suggest that P-His2 and this loop are critical to formation of holoCcmE, and that it also plays a role in release. Because R128$^C$ in the WWD domain (Fig. 2a) is near the heme pyrroles, we focused on its activity relationships. R128$^C$ variants showed that while lysine, and to a lesser extent, alanine, are functional (Table 1 and Supplementary Fig. 5), R128$^D$ shows no ability to form holoCcmE (or cyt $c$). This property is consistent with the vital role that the WWD domain plays in heme binding to attach to CcmE. Later we further compare our CcmC WWD domain structure to the recently determined structure of CcsBA (which attaches heme to cyt $c$ CXXCH), yielding a remarkable structural conservation for the WWD domain in binding heme, and in a unifying theme for structure/function activities of the superfamily. For most of the mutants, heme-staining results showed that the holoCcmE expression level is comparable with that of the WT, thus these mutants have normal expression level of CcmE and CcmABCD. To test if the inactivity of holoCcmE production is caused by expression level, we expressed the WT and mutant proteins with His-tag on both CcmA and CcmE and did Western blot to test the expression level of the mutant. Our results showed that these mutants had normal expression level of CcmE and CcmA (Supplementary Fig. 3b).

## Conformational changes of CcmABCD bound with substrates: ABC transporter release mechanisms (of heme)

Three-dimensional classification in cryo-EM studies reveals two major states of apoCcmABCD and CcmABCD in complex with ATP and heme, and one state of CcmABCD complexed with AMP-PNP. In one of the apoCcmABCD conformations, each CcmA binds one inorganic phosphate and the two CcmABs tightly contact to form a "full-closed" conformation (Fig. 3a). In the other conformation, the inorganic phosphate dissociates and CcmABs form a "widely open" inward-facing conformation (Fig. 3b). Only one major conformation is obtained for CcmABCD in complex with the ATP homolog AMP-PNP which shows a "full closed" inward-facing conformation (Fig. 3c). When both CcmA are bound with ATPs, CcmABs form the "full-closed" conformation which is very similar in the apoCcmABCD (Fig. 3d). The

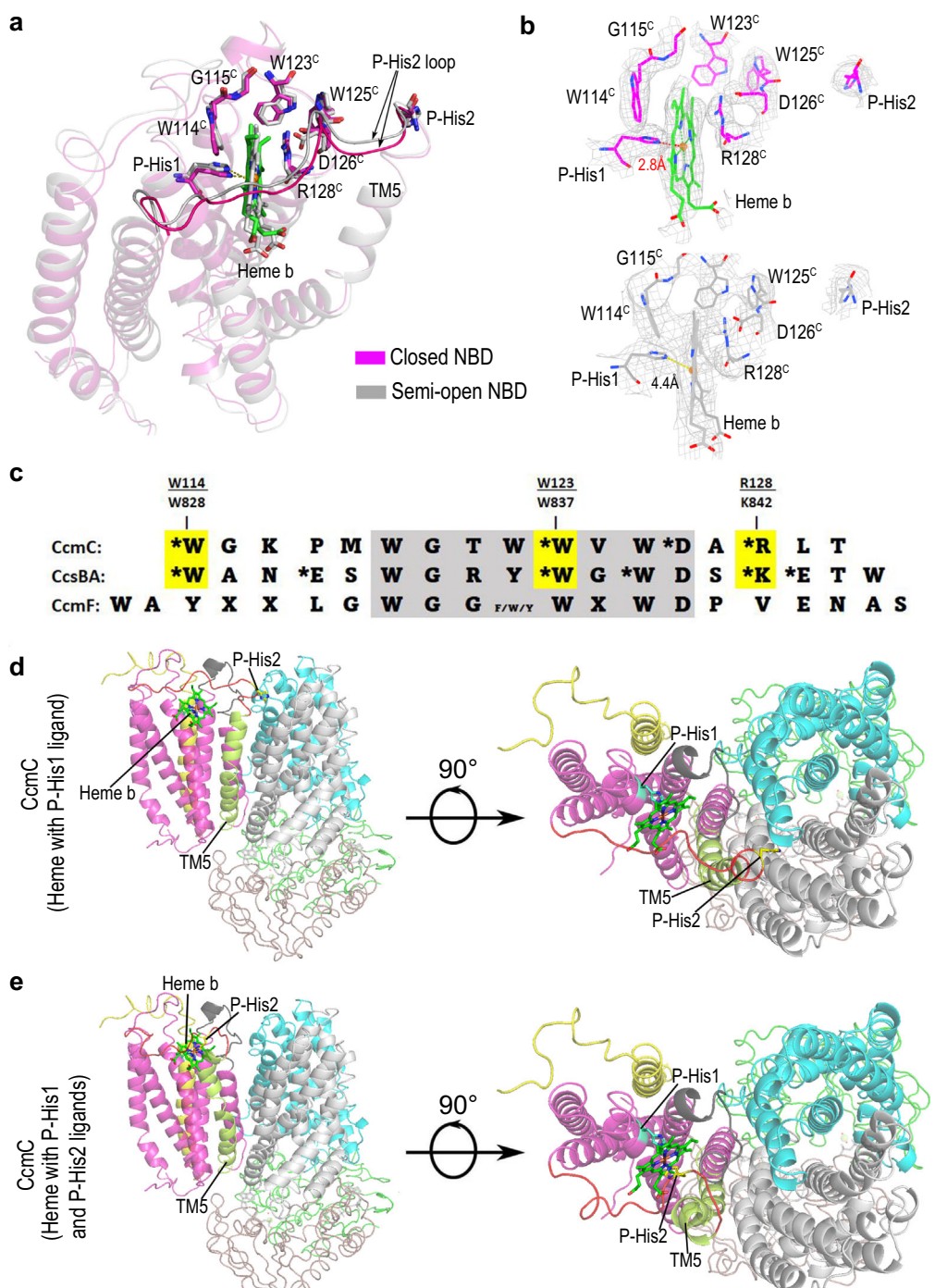

**Fig. 2 | Heme-bound structure and heme trafficking. a** Comparison of heme binding sites of closed NBD (magenta) and semi-open NBD (gray) conformations. In the closed NBD conformation, P-His1 is liganded to heme. In the semi-open NBD conformation, the distance and angle between the heme and P-His1 indicates that the heme is not liganded. **b** CryoEM density for bound heme and the surrounding residues in the closed NBD (upper) and semi-open NBD (lower) conformations. **c** The sequence alignment of the WWD domain of CcmC, CcsBA and CcmF. The indicated residues are from CcmC (e.g., W114 of *E. coli*) or CcsBA (e.g., W828 of *Helicobacter hepaticus*). Residues that biochemically crosslink to heme when substituted for cysteines have asterisks. **d** CcmABCD structure with heme liganded by only the P-His1 ligand with the P-His2 loop shown in red, shown from a side view and top down view. **e** CcmABCD structure with heme liganded by both the P-His1 and P-His2 ligands with P-His2 loop shown in red, shown from a side view and top down view, the structure modeled as described in Methods.

heme in the heme-binding pocket of CcmC is coordinated with H60$^C$ (P-His1), and in close contact with a distant R128$^C$ (Fig. 2a). H184$^C$ (P-His2), which is located right before the flexible loop and was previously proposed to coordinate with the heme iron together with H60$^C$, is too distant from the heme to coordinate with the heme (Fig. 2a). A conformation with only one ATP bound, near CcmC, exhibits a "semi-open" inward-facing conformation. In this semi-open state, heme is

loosely associated in the CcmC binding site and does not form a coordination bond with P-His1 (Fig. 2a, b). In all of the structures, the P-His2 loop of CcmC (residues W180$^C$-R199$^C$) is highly flexible, indicated by low density in the maps.

Of the different conformations, CcmABCD complex shows obvious relative shift among the subunits. Superposition of each of the closed NBD states of CcmABCD indicate nearly identical structures

**Table 1 | In vivo activities of mutants**

|  |  | cyt *c*-553 synthase function (%) (mean ± s.d.[a]) | holoCcmE production |
|---|---|---|---|
|  | WT | 100% | Yes |
|  | Negative control | 0% | No |
| CcmA mutants | N36A[A] | 0% | Yes |
|  | K40A[A] | 0% | Yes |
|  | T41A[A] | 0% | Yes |
|  | H81A[A] | 82.7% ± 5.7% | Yes |
|  | D151A[A] | 0% | Yes |
|  | H184A[A] | 0% | Yes |
| CcmB mutants | H17A[B] | 61.9% ± 16.5% | Yes |
|  | F27H[B] | 46.7% ± 14.3% | Yes |
|  | F70A[B] | 47.3% ± 7.3% | Yes |
|  | D73A[B] | 58.9% ± 7.4% | Yes |
|  | D76A[B] | 39.8% ± 4.9% | Yes |
|  | L79A[B] | 19.2% ± 4.4% | Yes |
|  | H100A[B] | 46.7% ± 13.5% | Yes |
|  | H100Y[B] | 50.5% ± 6.5% | Yes |
|  | H186A[B] | 52.5% ± 15.4% | Yes |
| CcmC mutants | H60A[C] | 0% | No |
|  | R128A[C] | 31.1% ± 8.2% | low |
|  | R128D[C] | 0% | No |
|  | R128K[C] | 79.2% ± 9.0% | Yes |
|  | H184A[C] | 0% | No |
|  | H147A[C] | 58.0% ± 7.4% | Yes |
|  | H174A[C] | 55.1% ± 10.1% | Yes |
|  | ΔG186[C] | 56.8% ± 18.9% | Yes |
|  | G186A[C] | 74.6% ± 4.3% | Yes |
|  | ΔS187[C] | 0% | low |
|  | S187A[C] | 65.9% ± 14.0% | Yes |

[a]Data are represented as mean ± s.d. from of *n* = 3, where '*n*' represents number of independent experiments. Source data are provided in the Source Data file.

(e.g., with two ATPs or two inorganic phosphates or two AMP-PNPs (Fig. 4a, b)). Compared with the closed NBD state, both $CcmB_2$ and CcmC in the open NBD state move relative to $CcmB_1$, indicated by the displaced CcmC and $CcmB_2$ subunits (Fig. 4c). The semi-open NBD state shows a similar but smaller conformational change as the open NBD state (Fig. 4d). More importantly, CcmC undergoes significant conformational change during closed NBD state to open NBD state transition. CcmC-TM4 predominantly interacts with $CcmB_1$-TM1 and $CcmB_2$-TM5 and CcmC-TM5 predominantly interacts with $CcmB_2$-TM4 and $CcmB_2$-TM5 (Fig. 4e). Particularly noteworthy between open and closed NBD states is movement between CcmC TM4 and TM5 which interact tightly with CcmBs (see oval highlight in Fig. 4c). This interaction leads to a significant conformational change of the whole CcmC structure. When CcmCs of the open NBD state and close NBD state are aligned with CcmC TM4, although the CcmC TM5 shows minor movement, there is a significant conformational change of the loop between TM4 and TM5. Other helixes and loops in CcmC also show large displacement (Fig. 4f), indicating a significant local conformational change of CcmC that may cause heme to form and break coordination with P-His1.

To investigate the roles of the key residues in CcmA and CcmB that may affect the formation of holo-cyt *c* and/or holoCcmE, we performed in vivo activity studies as described above. For CcmA interrogation, N36A[A], K40A[A], T41A[A], H81A[A], D151A[A] and H184A[A] were each mutated to alanine. H81A[A] and T41A[A] are coordinated with the $Mg^{2+}$ in the

ATP binding site, while N36A[A], K40A[A], D151A[A] and H184A[A] are in close contact of ATP. All CcmA variants, except H81A[A], abolished the production of holo-cyt *c*. Importantly, all retained the production of holoCcmE (Table 1 and Supplementary Fig. 5). As we have shown previously with a selected ATP-hydrolysis variant of CcmA[21], the holoCcmE in these variants are not released from the CcmABCDE complexes. In vitro ATPase assays show that, all ATP binding site mutants, except H81A, have low ATPase activities compared to WT (Supplementary Table 1 and Supplementary Fig. 3d). Thus, the conformational changes during hydrolysis and the ATPase cycle are essential for release. For CcmB variants (Table 1 and Supplementary Fig. 5), all produce holoCcmE (like CcmA variants) and all show some cyt *c* synthesis. We suggest that these results stem from multiple sites of interaction between CcmB and CcmC (and CcmA), thus a redundancy in the energy conversion/conformational-based release. CcmB is a conduit to CcmC to induce the ATP-dependent conformational states.

**A consensus model for heme binding and release in the WWD/P-His site of the WWD protein family**

As mentioned above, CcmC has a hydrophobic tryptophan-rich motif (WWD domain) of which at least two tryptophan residues W114[C] and W119[C] are in close contact with the heme in both heme-bound conformations (Fig. 2a, b). It has been demonstrated previously that specific residues in the WWD domain, when changed to cysteines, crosslink to heme (vinyls) in CcmCDE[26], establishing the WWD domain as part of the heme binding site (see Fig. 2c, crosslinked residues have asterisks). The same type of crosslinking interrogation was done for the WWD family of CcsBA[28], the System II cyt *c* synthase and heme transporter (Fig. 2c). Our CcmABCD structure containing heme in the closed NBD conformation, liganded by P-His1, allows us to make comparisons to CcsBA structure, and to evaluate the heme crosslinks (to vinyls) that have been published for both CcmC and CcsBA (Fig. 5). We focus on three residues that crosslink in both CcmC and CcsBA (W114[C]/W828; W123[C]/W837; R128[C]/K842, respectively CcmC/CcsBA). The conserved structural features of the WWD domain and P-His1 in both proteins and their interaction with heme are striking (Fig. 5a–c). Just as noteworthy are the locations of the three major common crosslinks in CcmC (Fig. 5a), CcsBA (Fig. 5b) and overlayed with the mutated cysteines that crosslink (Fig. 5c). The W114[C] crosslink was predicted to crosslink to vinyl 4 of heme because it could be double crosslinked with the His130[E] covalent linkage[26]. That is, W114[C] was crosslinked to vinyl 4 and His130[E] to vinyl 2, forming a large CcmC:heme:CcmE molecule, connected by covalently attached heme. Because His130[E] has been preliminarily assigned[29] to covalently attach to vinyl 2 (by resonance Raman spectroscopy), W114[C] was preliminarily assigned to vinyl 4. The structure demonstrates the close proximity of W114[C] (and W828 in CcsBA) to vinyl 4, a result that clearly indicates His130[E] must be attached to vinyl 2. The WWD domain interacts with the heme on the edge with vinyls, where the apoCcmE substrate will be covalently attached to His130[E]. His130[E] is present in a HXXXY motif, where Tyr134[E] will become the heme ligand in released holoCcmE[29]. For CcsBA, the substrate is CXXCH, where the histidine becomes the axial ligand upon release. One other parallel is that P-His2 is proposed to exchange for the ultimate axial ligands of products (Tyr134[E], and histidine of the CXXCH cyt *c*). Thus, a common feature is not only the WWD structural conservation, but unifying mechanisms of release and axial liganding of products (see "Discussion").

Sequence conservation of the WWD domain defines in part this family of proteins (CcmC, CcsBA, and CcmF)[15,19]. Sometimes called the "heme handling family"[30], all three families are involved in attaching heme to an acceptor (CcsBA and CcmF/H to CXXCH, and CcmC to apoCcmE, or HXXXY). Now, the structures of CcmC and CcsBA WWD domains illucidate the molecular basis for conservation, and can be used as the foundation to understand functions of specific residues in the context of heme handling. Although a structure of CcmF is known,

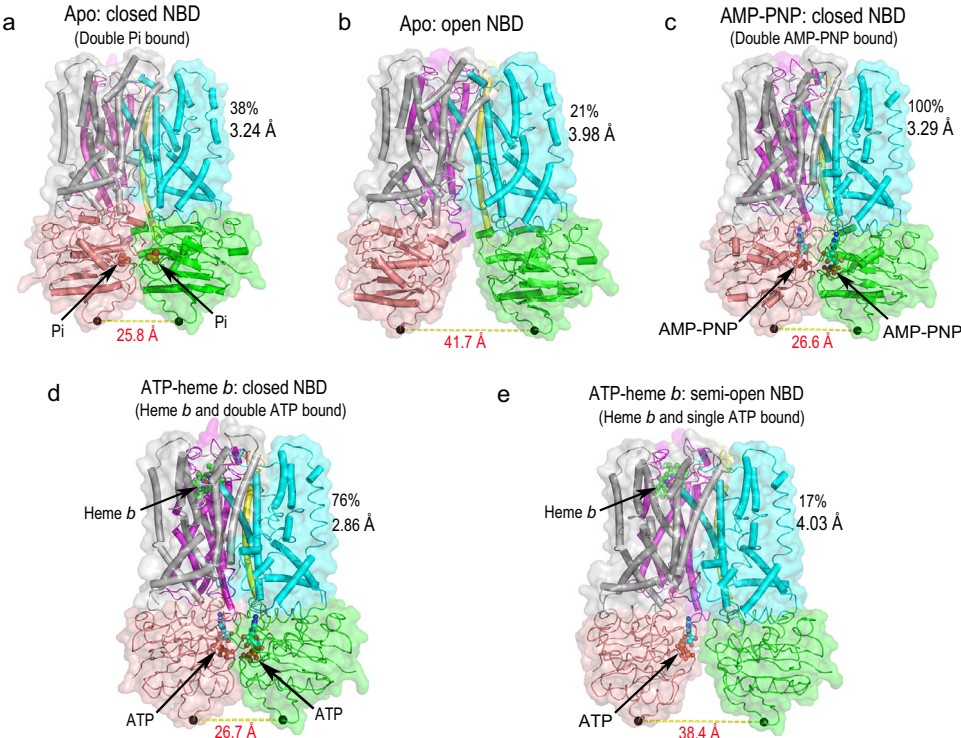

**Fig. 3 | The CcmABCD complex undergoes conformational changes upon the binding of substrates. a** Closed NBD conformation of CcmABCD bound with two phosphates. **b** Open NBD conformation of ligand-free CcmABCD. **c** Closed NBD conformation of CcmABCD bound with two AMP-PNPs. **d** Closed NBD conformation of CcmABCD bound with two ATPs and one heme. **e** Semi-open NBD conformation of CcmABCD bound with one ATP and one heme. The distance between the two Ala190s (shown in black spheres) on each NBD is indicated by yellow dash line and labeled in red font.

the CcmF/H cyt *c* synthase and the location of heme in the P-heme site still remains a mystery[22]. Nevertheless, the CcmC structure here with exogenous heme, and the CcsBA structure with endogenous heme in the P-heme sites (WWD/P-His ligands)[27], facilitate predictions on the roles played. Clearly the basis for heme binding by the WWD domain is explained, with almost half of the heme molecule showing close contacts (Figs. 2a, 5a–c). Moreover, the proximity of key residues in the domain (e.g., W114[C], W123[C], R128[C], L129[C]) to vinyls suggest that these sidechains may assist the heme attachment process. For CcmC, this means that the incoming His130[E] may be guided to the 2 vinyl, possibly even assisting adduct formation. We have previously discussed the chemistry behind His130[E] attachment to the beta carbon of vinyl 2, where oxidized heme (Fe$^{+3}$) would be favored[15,25]. It is possible that for example, R128[C] assists this chemistry.

Finally, all three family members must release the covalently attached heme products. In the case of CcmC, this release demands a conformational change catalyzed by the CcmABCD release complex. Once release is completed, the WWD domain in the context of CcmCD must reset for another round of heme binding, attachment to CcmE, and release. The CcmD peptide, with a single TM, has a conserved periplasmic residue (Tyr17[D]) that coevolved with the WWD residue Met118[C], and in theoretical structures these were placed adjacent[26]. Here we show that in the experimental structures, they are adjacent. We speculate that CcmD may aid in resetting the WWD domain for another round of heme/CcmE binding, once the CcmABCD-mediated release of holoCcmE is complete. We address in the Discussion how heme might traffick to the P-heme site in CcmC, comprised of the WWD/P-His domain.

## Discussion

Step 1 of the System I cyt *c* maturation pathway is catalyzed by the ABC transporter release complex CcmABCD. Our cryo-EM structures reveal that CcmA$_2$B$_2$ form a homodimer which is an ABC transporter analog.

CcmC which harbors the heme-binding site is attached to two CcmBs with two coupling helices TM4 and TM5 on CcmC, respectively. A small single TM protein CcmD is bound to CcmC only. Our structures also reveal substantial conformational changes (Fig. 3) upon binding of ATP (to CcmA) and heme (to CcmC): closed NBD conformation when bound with two inorganic phosphates, two ATPs or two AMP-PNPs; open NBD conformation when it is ligand-free; and semi-open NBD when bound with one ATP. All CcmA variants except H81A[A] had low ATPase activities compared to WT. H81A[A] is also the only CcmA variant that showed WT levels of cyt *c* and holoCcmE biosynthesis. Thus, these results support the contention that ATP hydrolysis is required for release of holoCcmE (heme) and cyt *c* biogenesis. Although N36A[A] showed approximately 70% ATPase activity, it does not release holoCcmE or synthesize cyt *c*. N36A[A] is located at the interface of two CcmAs. In the closed-NBD state of apoCcmABCD bound with inorganic phosphates, N36A[A] is H-bonded with the phosphate, and A156[A], G130[A] in the other CcmA (Supplementary Fig. 2a, right zoomed in panel). One possibility is that N36A[A] blocks the exit path of the phosphate. For WT CcmABCD, the protein changes to open-NBD state to break these H-bonds and release the phosphate for the next cycle. But it is possible that for the N36A[A] variant, the phosphate leaks in the closed-NBD state and the next cycle of ATP hydrolysis starts without changing to open-NBD state. Thus heme cannot be released to produce holo-cyt *c*. The ABC transporter CcmA$_2$B$_2$ has proved not essential for heme transportation to CcmE because the CcmAB genes deletion does not affect holoCcmE formation[24,25]. Genetic analyses of more CcmABC variants here confirms that only mutations in CcmC but not in CcmA or CcmB, prevent holoCcmE formation (Supplementary Fig. 5). We propose that CcmA$_2$B$_2$ serves as a mechanical lever to transduces conformational changes from NBDs to CcmB TMDs (transmembrane domains) and further to CcmC via the two coupling helices TM4 and TM5 on CcmC (Fig. 4e). When CcmA$_2$B$_2$ is in the closed NBD conformation, the CcmC P-His1 is enabled as an axial ligand to the heme, which, with P-His2

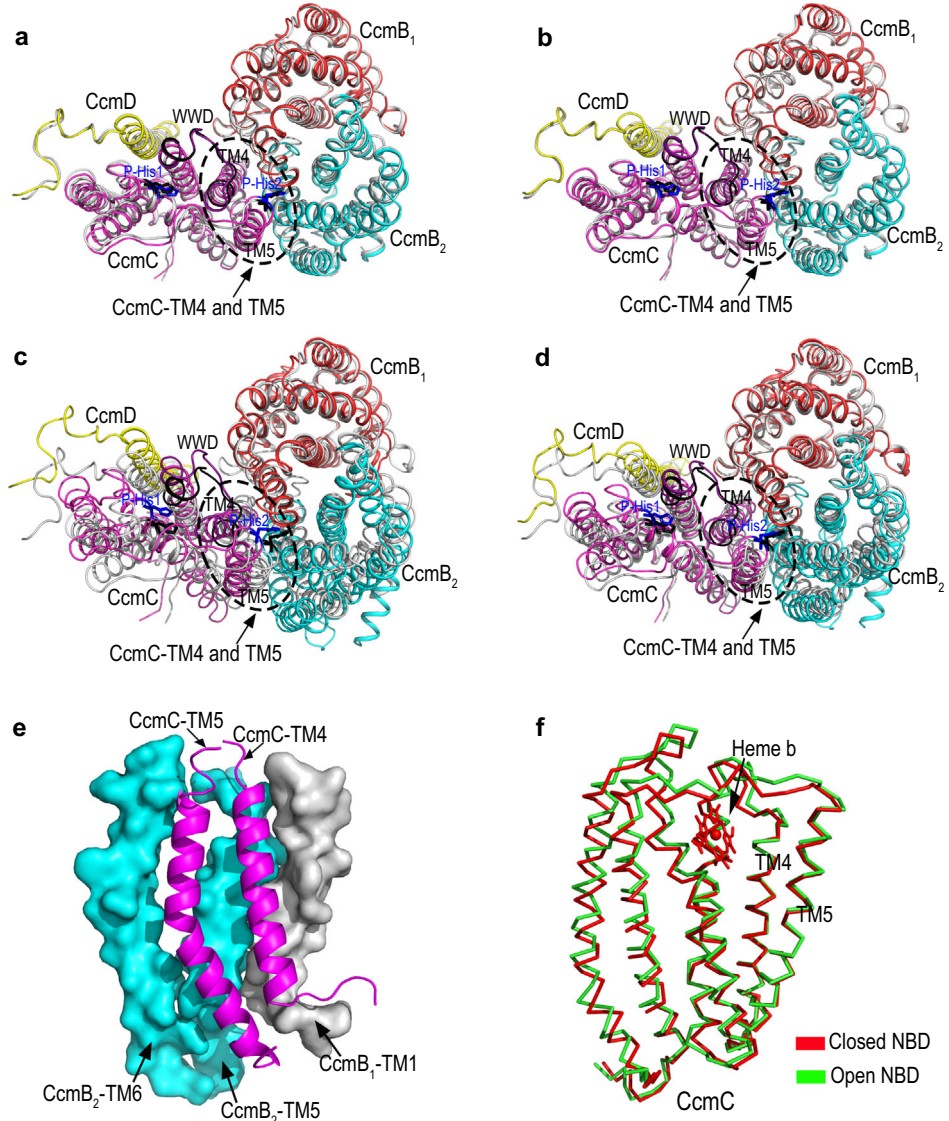

**Fig. 4 | Superposition of the closed NBD state of CcmABCD bound with two ATPs with other states.** The two structures of each panel (**a**–**d**) are aligned with CcmB$_1$. In each panel, P-His1 and P-His2 of closed NBD state are represented in black sticks, and in blue sticks for other states. The regions of CcmC-TM4, TM5, and P-His2 are marked by elipse. **a** CcmABCD closed NBD state bound with two ATPs (gray) and closed NBD state bound with two inorganic phosphates (colored). **b** CcmABCD closed NBD state bound with two ATPs (gray) and closed NBD state bound with two AMP-PNPs (colored). **c** CcmABCD closed NBD state bound with two ATPs (gray) and ligand-free open NBD state (colored). There is a significant conformational change in the CcmC-TM4 and TM5 region. **d** CcmABCD closed NBD state bound with two ATPs (gray) and semi-open NBD state bound with one ATP (colored). **e** CcmC-TM4 predominantly interacts with CcmB$_1$-TM1 and CcmB$_2$-TM5, and CcmC-TM5 predominantly interacts with CcmB$_2$-TM5 and CcmB$_2$-TM6. **f** To investigate the local conformational change of CcmC during closed NBD state (red) to open NBD state (green) transition, the two structures are aligned with CcmC-TM4. TM5 shows relatively minor movement related to TM4, but for the rest of CcmC including the heme binding site, there is a significant local conformational change as indicated by the large displacement of the helixes and loops.

ligeanding, facilitates the formation of holoCcmE (Fig. 4a, b). When CcmA$_2$B$_2$ is in the open NBD state (Fig. 4c), the two CcmBs move away from each other and causes a relative movement between CcmC TM4 and TM5, leading to a significant conformational change in the heme-binding site (Fig. 4f) that breaks the bond between P-His1 and heme, and subsequently allows the holoCcmE to release. An important feature of this release hypothesis involves the flexible P-His2 loop, mediated by CcmC TM5 (Fig. 2d, e). Based on our structures, together with our previous biochemical studies, we propose a mechanism for the ABC transporter release complex CcmABCD (Fig. 6 and Supplementary Movie 1).

CcmABCD is a unique ABC transporter from both architectural and functional perspectives. The tight binding of heme in the CcmCDE active site requires release of the heme (as holoCcmE) to complete the folding of CcmE, including formation of the Tyr134$^E$ iron ligand. The

released heme in the periplasmic CcmE chaperone is then used in the final cyt *c* biogenesis step (Supplementary Fig. 1). The transport of its substrate (heme) to the CcmC periplasmic binding site is not part of the ATP-dependent mechanism, distinguishing CcmABCD from other ABC transporters[31–35]. The architecture and stoichiometry also are exceptional. Perhaps the best functional analogy is to another ABC transporter, the LolCDE transporter in Gram negative bacteria for which structures were recently determined[36,37]. LolCDE binds to lipo-proteins that require transfer to the outer membrane from the inner membrane. The LolCDE protein complex translocates lipoprotein from the inner membrane outer leaflet to the chaperone protein LolA in the periplasm. Once released from LolCDE, LolA shuttles the lipo-protein to the outer membrane lipoprotein LolB for integration into the outer membrane. The LolCDE protein forms an ABC transporter composed of a heterodimer of TMDs LolE and LolC and a homodimer

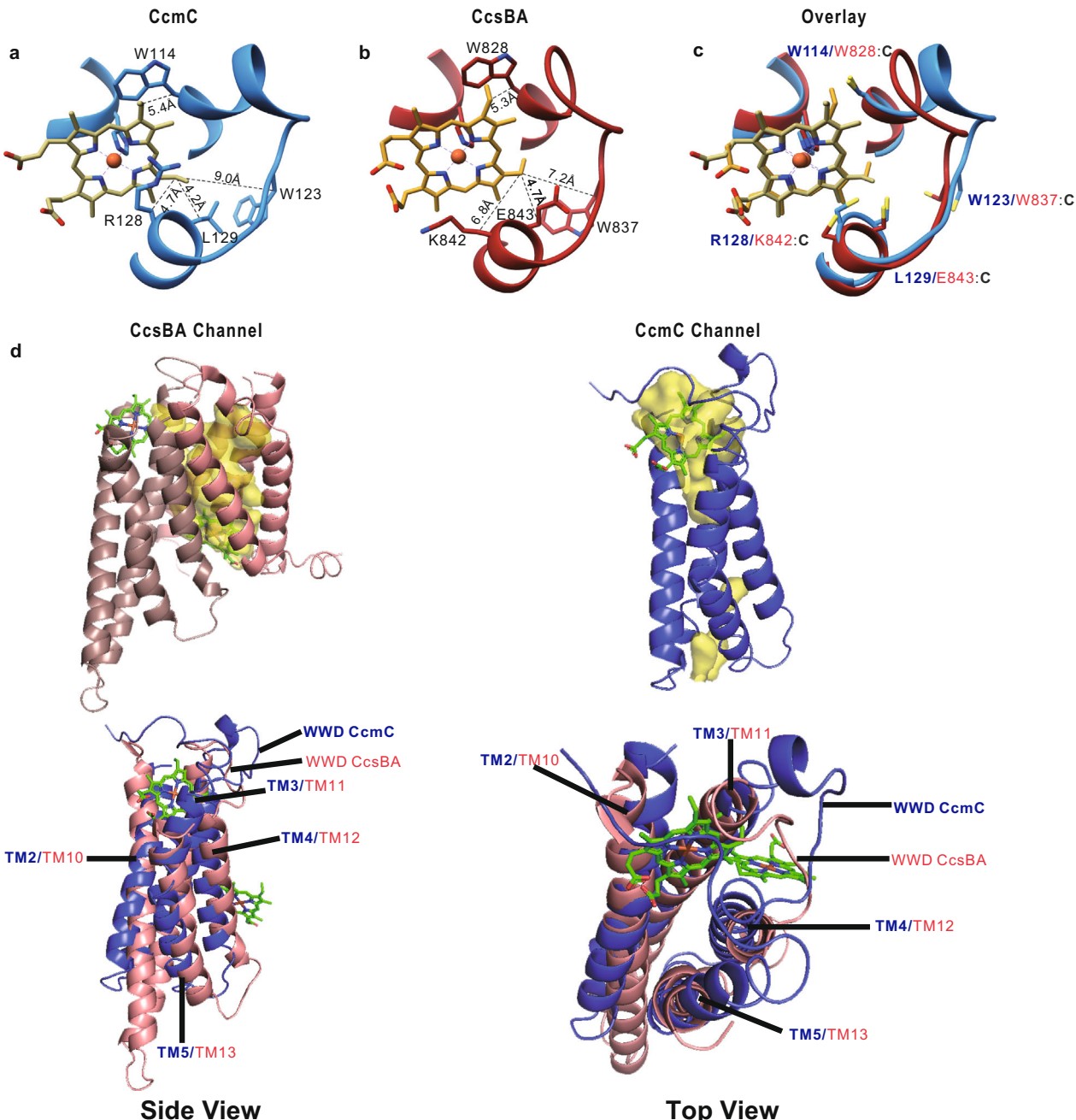

**Fig. 5 | Selected views of CcmC and CcsBA structures displaying WWD domain and potential heme channels. a** CcmC WWD domain and heme with residues known to crosslink to heme shown with their distances to respective heme vinyl groups labeled. **b** CcsBA WWD domain and heme with residues known to crosslink to heme shown with their distances to respective heme vinyl groups labeled. **c** CcmC (blue) and CcsBA (red) WWD domain overlayed upon one another surrounding heme from CcmC with residues known to crosslink to heme mutated to Cys. **d** CcsBA (PDB: 7S9Z) channel shown in yellow with potential channel in CcmC shown based off of location of CcsBA channel. Bottom half of D shows the two hemes of CcsBA with an overlay of CcsBA (pink) and CcmC (blue) to show a potential path heme could take into CcsBA.

of NBDs LolD (Supplementary Fig. 6). The lipoprotein is selected by and bound to a central channel of the transmembrane domains, translocated to LolA from LolC, and then the LolA-lipoprotein complex is released. The process requires ATP binding and hydrolysis. ATP binding causes the closed NBD conformation and changes conformation of LolC to bring the bound lipoprotein above the TMDs interface. ATP hydrolysis induces the open-NBD conformation and makes lipoprotein accessible to LolA, subsequently transferring it to LolA, followed by LolA-lipoprotein complex dissociation.

Architecturally, CcmABCD and LolCDE are distinct (Supplementary Fig. 6). In LolCDE the substrate lipoprotein is bound to a V-shaped channel formed by LolC and E; whereas for CcmABCD the heme-binding site is on the CcmC subunit, which is attached to the ABC transporter CcmA$_2$B$_2$. The conformational changes of both LolCDE and CcmABCD coordinate with substrate binding in the binding sites, substrate attaching to the shuttle protein LolA or CcmE, and the release of shuttle protein-substrate complex. Very recently, a theoretical structure of CcmABCD was published as a working example for a powerful neural network predictor (AF2complex) of multimeric protein complexes[38]. Their predicted models of CcmABCD to our experimental structures are very similar, ranging from 0.82 to 0.89 TM scores for overall structures. As one might expect, the substrate

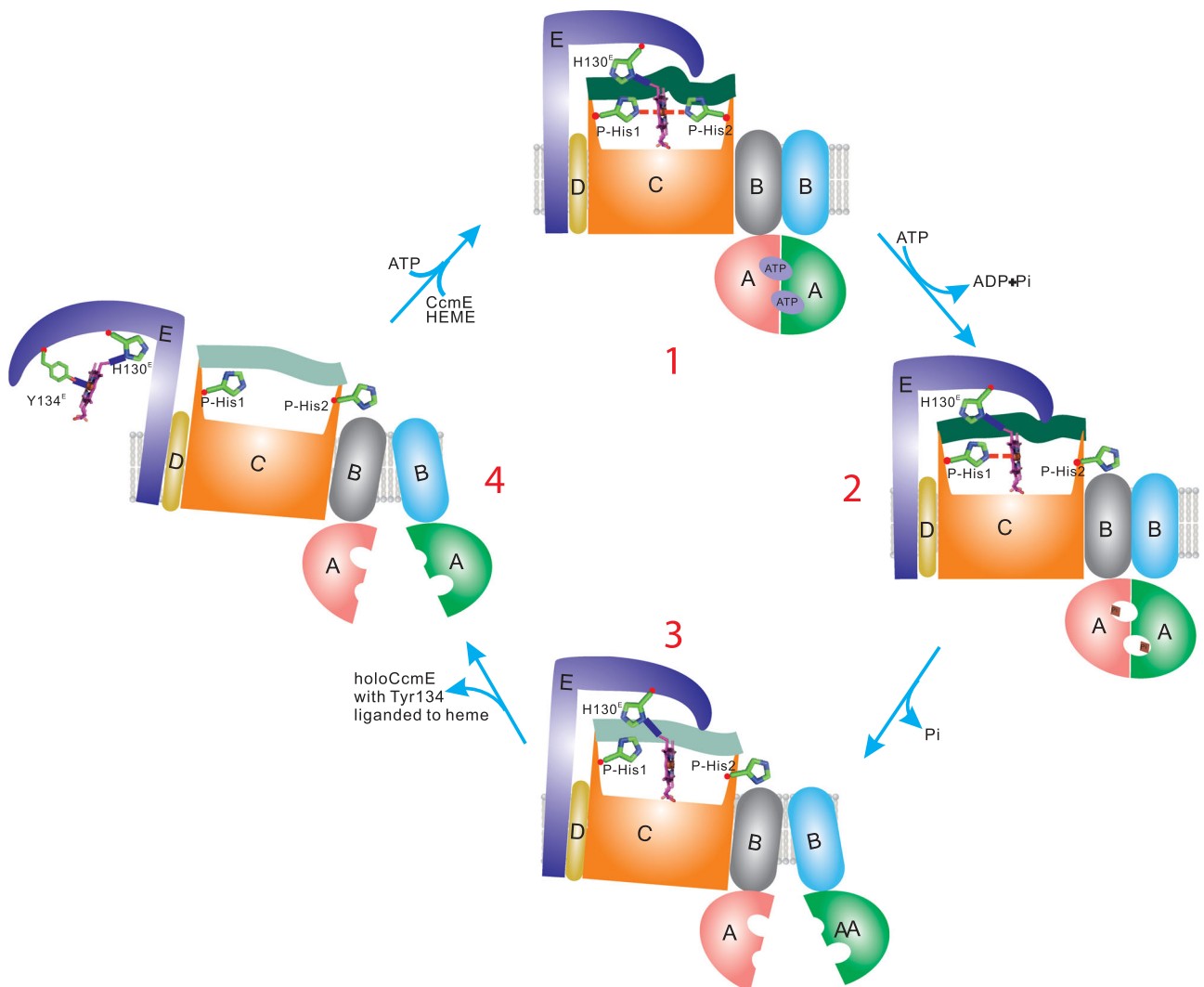

**Fig. 6 | Cycle of CcmABCD structures and functions.** The general function of CcmABCD is to attach heme to the periplasmic heme chaperone, CcmE, in the System I cyt *c* maturation pathway. Structures presented here provide the molecular basis for this function. The hypothetical cycle begins (#1) with CcmABCD containing all substrates: heme liganded by CcmC P-His1 and P-His2, the His130$^E$ adduct to vinyl 2 of heme, and two ATP molecules bound by a CcmA dimer in the closed NBD state. The structures represented by #1 to #4 are discussed in the text, whereby hydrolysis of ATP leads to the CcmB:CcmC interaction that pulls P-His2 (and the CcmC outer TM5) towards CcmB$_1$. This in turn allows the Tyr134$^E$ to ligand the heme iron (i.e., a P-His2 to Tyr134$^E$ ligand switch) and release as holoCcmE (see #4). Note that the conformational changes induced by ATP hydrolysis that lead to major movements of CcmC P-His2/TM5 could occur at #2 or #3. See Supplementary Movie 1 for a 30 s representation using the structures for the cycle and ATPase-based release of holoCcmE.

binding site with heme in the theoretical models do not provide enough accuracy to prove heme orientations, heme vinyl group additions, or the mechanisms for heme ligand switching (CcmC histidines for CcmE tyrosine). Our structures demonstrate the absolute need for experimental structures with multiple conformations. Additionally, Cryo-EM structures with and without substrates (heme and ATP) facilitate an understanding of the dynamic heme transfer and release processes.

A question that remains for CcmC is how heme traffics to the WWD/P-His external binding site in CcmC. It is likely that binding of apoCcmE to CcmC causes a conformational change[39] that allows heme access, for liganding by P-His1 and P-His2[25,26]. We envision one of two paths for heme transport: heme could enter from below (cytoplasmic entry) through a channel. Alternately, CcsBA transports heme bound to two TM histidines (called TM-His1 and TM-His2) to the external WWD/P-His site through a channel (Fig. 5d)[27,40]. However, CcmC does not possess TM histidines, and a channel is not as clear (Fig. 5d). Various views on CcmC compared to CcsBA suggest that heme could come from the outer leaflet or a channel that opens, for example, upon

CcmE binding (Fig. 5d). Because heme in CcmC is proposed to require oxidation (to Fe$^{+3}$) for His130$^E$ adduct formation[15], it likely does not need protection from oxidation like in CcsBA[40,41]. In CcsBA, reduced heme (Fe$^{+2}$) is needed for thioether formation to CXXCH. Thus, whether heme enters CcmC through a channel from below or from the outer leaflet will require further studies.

The superfamily of membrane proteins containing the WWD domain have three family members (CcsBA, CcmC, CcmF), of which two (CcsBA, CcmF) have structures that were recently determined[22,27]. Both CcsBA and CcmF/H are cyt *c* synthases that attach heme to cyt *c* (a CXXCH motif). CcmC is distinct in that it attaches heme to a heme chaperone called CcmE, and is the only member that is a component of an ABC transporter (CcmABCD). After the heme is attached in all three members, the heme product (cyt *c* or holoCcmE) must be released for subsequent folding. The mechanisms of release of cyt *c* from CcmF and CcsBA remain elusive, but our structures for CcmABCD elucidate the structural basis behind the release mechanisms for holoCcmE. The highly conserved WWD domain (Fig. 2c), in combination with P-His1 and P-His2 iron ligands, binds heme. We show that the remarkable

structural homology of the WWD domain between CcsBA and CcmC explains its sequence conservation in the context of heme handling. For the CcmCDE heme complex, an exchange of two strong histidine ligands with a weak Tyr134$^E$ and small ligand (e.g., water), requires a major disruption of the heme binding site. It is clear from the various conformations of CcmABCD (with and without substrates), that dislodging the P-His1 and P-His2 ligands are the basis for the ligand switching, holoCcmE release, folding, and thus ability to chaperone heme to the CcmF/H synthase (Supplementary Fig. 1). This ATP-dependent mechanism of release and trafficking in system I may be the cost for the use of heme at very low concentrations for cyt $c$ synthesis, when compared to the CcsBA (system II) pathway[42,43]. We note that in all three family members, the flexible P-His2 loop, shown by its poor densities, is also a common feature of the synthesis, ligand switch, and release mechanism. The presence of this superfamily in bacteria, archae, and eukaryotes suggests these mechanisms are ancient and efficient in the trafficking and processing of heme.

## Methods

### Purification of cytochrome $c$ maturase complex CcmABCD

The construct containing cytochrome $c$ maturase complex CcmABCDE genes was generated by PCR from the *E. coli* BL21 (DE3) genome, and a hexahistidine tag was added at the N terminus of CcmA. The primers used in this study are listed in the Supplementary Table 2. The gene was subcloned into the expression vector of pET-28a(+). The recombinant CcmABCD protein was expressed in *E. coli* BL21 (DE3) strain grown at 23 °C in Luria–Bertani media after induction with 0.5 mM IPTG. Bacteria were collected and resuspended in 50 mM $KH_2PO_4$ (pH 8.0), 1 mM EDTA and a tablet of complete protease inhibitor (Sigma). This suspension was passed five times through a microfluidizer at a pressure of 15,000 psi to disrupt the cells. The disrupted cells were centrifuged at $20,000 \times g$ for 20 min to remove cell debris. Membranes were obtained after centrifugation at $180,000 \times g$ for 4 h and stored at −80 °C. On the day of purification, the membrane pellet was resuspended in 50 mM potassium phosphate buffer (pH 8.0) and solubilized with 1% DDM on ice for 1 h. Insoluble material was removed by centrifugation at $180,000 \times g$ for 20 min. The solubilized fraction was loaded onto an Ni-nitriloacetic acid (NTA) (Qiagen) column pre-equilibrated with 50 mM potassium phosphate buffer and 0.1% DDM (pH 8.0). Proteins were eluted with 250 mM imidazole in the same buffer. The peak fractions from the Ni-NTA column were pooled, concentrated, and loaded onto an ENrichTM SEC 650 10 × 300 Column (Bio-Rad, Cat# 7801650) equilibrated with gel filtration buffer containing 20 mM Tris-HCl (pH 7.5), 50 mM NaCl and 0.03% DDM. The purest and the most concentrated fractions were concentrated to 12.8 mg/ml using a Millipore 100 kDa cut-off concentrator and used immediately for cryo-EM grid preparation.

### Heme staining analysis

To investigate the roles of the key residues that affect the formation of holo-cyt $c$ and holoCcmE, we performed heme staining studies for the cell lysate of the *E. coli* strain BL21 with the CcmA-H genes knocked out (BL21-ΔCcm) complemented with a constitutive plasmid incorporated with CcmA-H gene containing different point mutations and another inducible plasmid encoding the reporter gene *H. pylori* cytochrome *c*-553 (cyt *c*-553). Protein samples were prepared by mixing cell lysate with DTT-free 6xSDS loading buffer and separated with 15% SDS-PAGE. Prior to staining, a 6.3 mM TMBZ (3,3′,5,5′-tetramethylbenzidine) stock solution was made in methanol and three parts of TMBZ stock solution was mixed with 7 parts 0.25 M sodium acetate (pH 5.0) to produce the TMBZ staining solution. The SDS-PAGE gel was incubated in the TMBZ staining solution for 1 h, followed by the addition of 30% $H_2O_2$ to a final concentration of 0.3% (V/V), and incubated for another 30 min to develop color[44]. The amount of holo-cyt *c*-553 was semi-quantified by summing up the pixel greyscale values of the band using ImageJ[45].

### Western blot

The gene encoding CcmABCDE WT or mutant with hexahistidine tags added on both of the N terminus of CcmA and the C terminus of CcmE was subcloned into the expression vector of pET-28a(+). The recombinant WT and mutants were expressed in BL21-ΔCcm strain grown at 22 °C in Luria–Bertani media after induction with 0.2 mM IPTG for 1 h when $OD_{600}$ reached 0.6. One milliliter of cells of each strain were harvested and the pellets were lysed with 200 μL of SDS loading buffer. Then the supernatants were loaded and separated on a 15% SDS–PAGE gel. The resolved proteins were electrophoretically transferred to PVDF membrane and blocked with 5% fat-free dry milk in TBST for 2 h at room temperature. Then primary His-Tag polyclonal antibody (Immunoway, Cat# YM3204) was added in 5% BSA in the ratio of 1:5000 and incubated overnight at 4 °C and incubated with horse-radish peroxidase (HRP) conjugated AffiniPure Goat anti-Rabbit IgG (H+L) secondary antibody (1:5000 dilution with 5% BSA, Boster Biological Technology, Cat# BA1054) for 2 h at room temperature and washed three times before detection. The blot was developed with Tanon 5200 Chemiluminescent System.

### ATPase assay

The CcmABCDE-associated ATPase activity was measured by ADP-GloTM Kinase Assay kit (Promega, Cat# V6930). The assay was carried out in solid white 384-well multiplates in triplicate. Before use, the plate and various assay buffer were equilibrated to room temperature. The purified wild type CcmABCDE or mutants were quickly thawed and diluted to 0.26 mg/ml (2 μM) with Kinase Reaction Buffer (40 mM Tris, pH 7.5, 20 mM $MgCl_2$, 0.1 mg/ml BSA). Reactions were initiated by adding 1.6 μl of 1 mM $Mg^{2+}$ and ATP to all wells. At this point, each reaction contained 1 μM proteins and 100 μM ATP. The solutions were mixed by gently vortexing and swirling, and incubated for 30 min at room temperature. After the kinase reaction, an equal volume of ADP-Glo™ Reagent was added to terminate the kinase reaction and deplete the remaining ATP. The solutions were incubated at room temperature for 40 min. Then 32 μl of Kinase Detection Reagent was added to convert ADP to ATP and the newly synthesized ATP was measured using a luciferase/luciferin reaction. 20 μl of the sample was transferred into separate wells of the multiplates and incubated at room temperature for 60 min to allow luminescent signal to develop. Then the solid white 384-well multiplate was read on a luminometer (SpectraMax i3x, molecular devices, USA). Subsequent data analysis and plotting were done using OriginPro (https://www.originlab.com/).

### Cryo-EM sample preparation and data collection

Stock solutions of $MgCl_2$ and AMP-PNP were mixed with the CcmABCD protein solution to make a solution containing ~12 mg/ml CcmABCD, 3.5 mM $MgCl_2$ and 3.5 mM AMP-PNP; Stock solutions of $MgCl_2$, ATP and heme $b$ were mixed with CcmABCD protein solution to make a solution containing ~12 mg/ml CcmABCD, 3.5 mM $MgCl_2$, 3.5 mM ATP, 3.5 mM heme $b$ and 1 mM DTT. The mixtures were incubated on ice for ~30 min before the grid preparation. Then, 3 μl of the sample was applied to a freshly glow-discharged 300 mesh Quantifoil R 2/1 gold grid and blotted for 2.5–4 s using a blotting force of −4 at 8 °C and 100% humidity in a FEI Vitrobot Mark IV (Thermo Fisher). The apo protein sample was treated the same way. Grids were flash-frozen in liquid ethane and stored in liquid nitrogen until data collection. Automatic data collection was performed on a Titan Krios microscope (Thermo Fisher), equipped with a K3 direct-electron detector (Gatan) operating at 0.832Å per pixel in counting mode, using the SerialEM software package. A total dose of 64.75 e$^-$/Å$^2$ (apoCcmABCD and CcmABCD–ATP-Heme $b$) or 68.65 e$^-$/Å$^2$ (CcmABCD–AMP-PNP) was fractionated into 45 frames of 40 ms each. A 100 μm objective aperture was used. We collected 5060 movies for the apoCcmABCD dataset, 4660 movies for the CcmABCD–AMP-PNP dataset and 3857 movies for the CcmABCD–ATP-heme $b$ dataset. Detailed data

collection parameters are summarized in Supplementary Table 3. Automated data collection was performed using SerialEM[46].

## Data processing

Frame movement of each micrograph was corrected using MotionCor2[47]. Contrast transfer function (CTF) of each motion-corrected micrograph was estimated using Gctf[48]. For each dataset, a small number of particles were automatically picked by RELION[49] from 1000 binned micrographs to generate initial two-dimensional (2D) class averages which were subsequently used as the templates for automatic particle selection of the entire datasets by Gautomatch (https://www.mrc-lmb.cam.ac.uk/kzhang/). Total numbers of 3,400,733 particles from the apo protein dataset, 2,327,524 particles from the CcmABCD–AMP-PNP dataset and 2,079,706 particles from the CcmABCD–ATP-heme *b* dataset were picked.

Subsequent data processing was carried out in cryoSPARC[50] including particle extraction, 2D classification, ab-initio reconstruction, heterogeneous refinement, local refinement and per-particle CTF refinement. For the apoCcmABCD dataset, two final classes of 270,353 and 147,351 particles were used to reconstruct a 3.24 Å resolution map of the closed NBD state and a 3.98 Å resolution map of the open NBD state, respectively (Supplementary Fig. 7). For the CcmABCD–ATP-Heme *b* dataset, two final classes of 270,825 and 58,617 particles were used to reconstruct a 2.86 Å resolution map of the closed NBD state and a 4.03 Å resolution map of the semi-open NBD state, respectively (Supplementary Fig. 8). For the CcmABCD–AMP-PNP dataset, only one major class of 142,289 particles were used to reconstruct a 3.29 Å resolution map of the close NBD state (Supplementary Fig. 9).

## Model building, refinement, and validation

The atomic models were built manually using Coot[51]. First, AlphaFold[52] predicted model of each subunit was fitted into corresponding map as a rigid body using UCSF Chimera[53]. Then the fitted model was manually adjusted and refined to correct errors in local regions to best match the density maps using Coot. Magnesium ions, AMP-PNP, ATP, and Heme *b* were fitted manually. The final model was refined using *phenix.real_space_refine*[54] with geometric constraints.

Modeling CcmABCD with the P-His2 ligand (for Fig. 2d, e): The experimentally determined CcmABCD model with two ATP and heme was used (where P-His1 is a ligand) as starting point. Residues 164–197 of the CcmC P-His2 loop were rigidly replaced with the same residues of the *E.coli* AlphaFold[52] structure using the "replace fragment" functionality within WinCooT. His184[C] (P-His2) liganding is based on the CcsBA cryo-EM (open) structure for proper rotation and rotamer to the iron of heme. The ends of this region (Residues 164 and 197) used the WinCooT[51] real space refinement tool to ensure that the chains connected with the original CcmABCD model.

Figures were generated using UCSF ChimeraX[55] and Pymol (The PyMOL Molecular Graphics System, Schrödinger, LLC).

For Supplementary Movie 1 on the CcmABCD cycle, structures of CcmABCD in different states, ATP and ADP ligands, heme, and the HDENY peptide (residues 130–134) from *E.coli* CcmE were used. HDENY peptide was manually attached to heme using winCOOT in two different states, with positioning based on the well-studied His130[E]–2 vinyl adduct and Tyr134[E] ligand to heme. The movie was generated using Chimera animation, and morph conformation features were used to display the conformational changes.

## Reporting summary

Further information on research design is available in the Nature Research Reporting Summary linked to this article.

## Data availability

Source data are provided with this paper. The source data underlying Table 1, Supplementary Figs. 5b, 3c, d, and Supplementary Table 1 are provided as a Source Data file. Cryo-EM maps and atomic coordinates have been deposited in the Electron Microscopy Data Bank (EMDB) and the Protein Data Bank (PDB) under the accession codes EMD-31394 and 7F02 (apoCcmABCD bound with inorganic phosphates in closed NBD state), EMD-31956 and 7VFJ (ligand-free apoCcmABCD in open NBD state), EMD-31395 and 7F03 (CcmABCD bound with AMP-PNP in closed NBD state), EMD-31396 and 7F04 (CcmABCD bound with heme *b* and double ATPs in closed NBD state), EMD-31957 and 7VFP (CcmABCD bound with heme *b* and single ATP in semi-open NBD state). Any additional data supporting the findings of this study other than deposited data described previously are available from the authors on reasonable request. Source data are provided with this paper.

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

## Acknowledgements

All cryo-EM datasets were collected at the Yale University West Campus Cryo-Electron Microscopy Core. We would like to thank S.W. for technical support on microscopy; P.C. for kindly help for data processing. This work was supported by National Key Research and Development Program of China 2020YFA0509400 (to J.P.); the Priority Academic Program Development of Jiangsu Higher Education Institutions (Integration of Chinese and Western Medicine); the Natural Science Foundation of Jiangsu Province for Young Scientists BK20190806 (to B.L.); Jiangsu Key Discipline Fund for the 14th Five-Year Plan; Start-up funds from Yale University (to K.Z.). R.G.K. is supported by NIH GM47909 and a Washington University Danforth seed grant.

## Author contributions

K.Z. and J.Z. designed the project. J.L. prepared all cryo-EM samples, processed the cryo-EM data, and built the models assisted by L.H. under the supervision of K.Z. and J.Z. W.Z., D.M., and E.L. performed and analyzed biochemical characterization assisted by M.G., Y.L., K.Y., M.S., Y.Z., X.M., and B.L. All were involved in analyzing the results. R.G.K., J.Z., K.Z., and J.L. prepared the manuscript assisted by other co-authors.

## Competing interests

The authors declare no competing interests.
