## [Peer Review File · Nature Communications]

Structures of the CcmABCD Heme Release Complex at Multiple StatesREVIEWER COMMENTS

Reviewer #1 (Remarks to the Author):

Jiao Li et al report cryo-EM structures of CcmABCD, a bacterial maturase complex that is involved in cytochrome c assembly. CcmABCD contains an ABC transporter (CcmA2B2) and the transmembrane subunits CcmC and CcmD that bind and release heme during the assembly process. The structural data include cryo-EM structures of the complex in an apo form, bound to the nucleotide analog AMP-PNP, or bound to ATP and heme. The authors find conformational changes in the ABC transporter unit CcmA2B2 that are propagated to CcmC, where they might weaken heme binding and trigger the release of heme from CcmCD. The structures appear well determined and carefully analyzed, and the insight is certainly novel and nicely complements earlier studies of the CcmF component of the heme/cytochrome assembly line. The authors include some functional analysis that appear to be consistent with the structural studies and support some, but not all, mechanistic conclusions the authors make. We have the following concerns / comments.

Main comments:

- The functional analysis of mutants and wild type measures endpoints of the biosynthetic assembly pathway, making specific conclusions about the steps catalyzed by CcmABCD challenging. We believe ATPase assays should be conducted to obtain more direct functional insight.
- Over-interpretation / over-speculation with respect to mechanistic conclusions: Firstly, the nature or existence of a conformational change of CcmC P-his-2 induced by the CcmABCD complex in order to release heme is not clear from the data presented. Secondly, based on the discussion (page14, 314-317), NBD opening is expected to lead to conformational changes in CcmC, helping break the bond between P-His1 and heme, subsequently allowing the holoCcmE release. However, it is not stated clearly if it is ATP binding ATP hydrolysis that is required to weakens heme interaction with CcmC. At present, several questions about the molecular mechanism of heme release are unanswered. The authors should therefore tone down their mechanistic speculations. If the authors want to make specific mechanistic statements about the above-mentioned points of the heme binding, transfer and release process, they would have to add supporting experimental data or MD simulations.
- The EM density for bound Heme and the sidechain of surrounding residues in the reported structures (closed NBD and semi-open NBD conformations) are not shown (or not shown sufficiently well) in the current version of the manuscript. This is essential information to evaluate the structural analysis.

- Page5, line99-101: The authors report that CcmE was co-expressed with CcmABCD, but not observed in the SDS-PAGE gel. In essence, the authors report an experiment that did not work (complex containing CcmE did not form). It's unclear what the authors conclude from this finding. Also, it's not possible to evaluate the statement because no SDS-PAGE analysis was shown. In general, the biochemical characterization of the purified protein is somewhat meagre. We suggest that the authors add protein purification and SDS-PAGE gel data to the supplementary figures.

- Page7, line 136-145, Fig2e and Fig S3: It is reasonable to generate a CcmABCD model with the P-His2 ligand by referring to a homology model. However, any comparisons based on the “predicted model” are rather speculative and would need experimental data to support the mechanistic conclusions drawn (eg, functional data with mutation or another experimental structure with the P-His2 ligand). At present, there is not enough evidence to support the notion that the interactions of TM5 and P-His2 loop of CcmC with CcmB represent the structural basis for holoCcmE release during the ATPase cycle. The mutation experiments tested with Cyt c synthesis function assay and HoloCcmE formation assay can only confirm that H60 and H184 are Heme ligands, but not that the interactions of TM5 and P-His2 loop of CcmC with CcmB represent the structural basis for holoCcmE release during the ATPase cycle. The reported deletion of S187 does not directly support conformational changes.

- Fig. S4: Synthetase functions of mutants.

There is no expression control shown, for example in the form of a Western blot. Hence, the expression levels of these mutant might differ and it is not clear if the observed reduction in activity is due to decreased expression levels or functional deficiency.

- Page8 177-179: The authors state that in the apoCcmABCD closed conformations, each CcmA binds one inorganic phosphate: The EM density of bound inorganic phosphate is not shown to be evaluated by the reader / reviewer. This information should be provided.

- Fig. S5: ATP binding sites. Because there are so many labels in the figure, the EM density is difficult to evaluate. Based on the density displayed in Fig. S5, we are not convinced that Mg²⁺ is bound. The authors should make a clearer figure showing the density of Mg²⁺ and the bound nucleotides. Furthermore, in the discussion (page 14 lines 305-306), the author state that semi-open NBDs contain a single bound ATP. The relevant EM density should be clearly shown.

Minor comments

- In the Methods section, Page 18 line 419:

The gel filtration buffer containing 20 mM Tris-HCl (pH 7.5) and 50 mM NaCl (no detergent). Could the author confirm this information? Was there really no detergent in this buffer?

- A molecular movie showing the observed conformational changes (e.g. morphing fully open; semi-open and fully closed states) would be helpful.

- Fig 5e: CcmC TM helix labeling is confusing (differences between left and right panel).

- Page 13, lines 292-298. We feel this text is redundant.

Reviewer #2 (Remarks to the Author):

The article entitled "Structure and Mechanisms of the CcmABCD Heme Release Complex" by Li et al., describes the CryoEM structures of the CcmABCD complex in several different conformations. The complex was investigated with ATP, AMP-PNP, or alone and showed distinct conformations. These along with site directed mutagenesis and protein characterization provides a mechanism for the movement of heme through the CcmABCD complex to CcmE. Overall this provides new information on ATP hydrolysis by the CcmABCD complex for the movement of heme to CcmE. There are a few major concerns. First although the resolution of the CryoEM structure is high (lowest 2.86 Å), there are several claims made throughout the text that would require additional data, such as crystallographic or spectroscopic, to be supported. These include distances measured that are below that of the structural resolution of ligands in the proteins. It is possible that there is existing data to support these observations, but these references are missing from the text. Second, there are multiple formatting and grammatical mistakes. For formatting neither the figures nor the supplemental figures are listed sequentially within the text. Figure 3 is the last figure to be mentioned in the text while Figure 5 is mentioned third in the text. Likewise the supplemental figures proceed from S1 to S5, then S4 and finally S6, S7, and S8. There was no mention of figures S2 or S3 within the text. This makes the flow of the figures and text difficult to follow. Grammatical mistakes are outlined below. Last, there is lots of data within the paper that is buried and not well explained. Very few of the variants, specifically CcmB, are discussed. An additional sentence or two within specific sections would be useful in further describing the findings. This is particularly apparent for all the variants and makes it hard to understand some of the discussion

including lines 306-308. It is unclear how or why deletion of these subunits resulted in a loading of CcmE unless a reference to explain this was omitted. Overall, the manuscript with major revisions to improve the explanation for the data interpretation and how the data support the conclusions drawn.

Specific corrections

Page 8, line 179, “full-close” conformation should be “full-closed” conformation. This close to should be changed to closed throughout.

Page 10, lines 228-229, the parenthesis around (of the WWD protein family) should be removed.

Also within the section on page 11 and throughout the manuscript, continue to use the superscript with the subunit letter for all of the variants. This is very useful to the reader.

Page 12, line 278, Beta doesn't need to be capitalized

Page 16, line 357, the word that should be changed to the

Page 16, line 365, Add the word Alternately before CcsBA, to compare the two paths.

REVIEWER COMMENTS

Reviewer #1 (Remarks to the Author):

Jiao Li et al report cryo-EM structures of CcmABCD, a bacterial maturase complex that is involved in cytochrome *c* assembly. CcmABCD contains an ABC transporter (CcmA₂B₂) and the transmembrane subunits CcmC and CcmD that bind and release heme during the assembly process. The structural data include cryo-EM structures of the complex in an apo form, bound to the nucleotide analog AMP-PNP, or bound to ATP and heme. The authors find conformational changes in the ABC transporter unit CcmA₂B₂ that are propagated to CcmC, where they might weaken heme binding and trigger the release of heme from CcmCD. The structures appear well determined and carefully analyzed, and the insight is certainly novel and nicely complements earlier studies of the CcmF component of the heme/cytochrome assembly line. The authors include some functional analysis that appear to be consistent with the structural studies and support some, but not all, mechanistic conclusions the authors make. We have the following concerns / comments.

Main comments:

- The functional analysis of mutants and wild type measures endpoints of the biosynthetic assembly pathway, making specific conclusions about the steps catalyzed by CcmABCD challenging. We believe ATPase assays should be conducted to obtain more direct functional insight.

RESPONSE: We performed ATPase assays on purified CcmABCD complexes of key variants of CcmA (the ATP binding cassette protein). We have added the results in text, figure, and table. Please refer to Fig. S3d and table S1. All CcmA variants except H81A^A had low ATPase activities compared to WT. H81A^A is also the only CcmA variant that showed WT levels of *cyt c* and holoCcmE biosynthesis. Thus, these results support the contention that ATP hydrolysis is required for release of holoCcmE (heme) and *cyt c* biogenesis. We added one more author Xiuxiu Ma because she contributed to atpase assay.

Although N36A^A showed approximately 70% ATPase activity, it does not release holoCcmE or synthesize *cyt c*. N36^A is located at the interface of two CcmAs. In the closed-NBD state of apoCcmABCD bound with inorganic phosphates, N36^A is H-bonded with the phosphate, and A156, G130 in the other CcmA (Fig. S2a, see below). One possibility is that N36^A blocks the exit path of the phosphate. For WT CcmABCD, the protein changes to open-NBD state to break these H-bonds and release the phosphate for the next cycle. But it is possible that for the N36A^A variant, the phosphate leaks in the closed-NBD state and the next cycle of ATP hydrolysis starts without changing to open-NBD state. Thus, heme cannot be released to produce holo-*cyt c*.

- Over-interpretation / over-speculation with respect to mechanistic conclusions: Firstly, the nature or existence of a conformational change of CcmC P-his-2 induced by the CcmABCD complex in order to release heme is not clear from the data presented. Secondly, based on the discussion (page14, 314-317), NBD opening is expected to lead to conformational changes in CcmC, helping break the bond between P-His1 and heme, subsequently allowing the holoCcmE release. However, it is not stated clearly if it is ATP binding ATP hydrolysis that is required to weakens heme interaction with CcmC. At present, several questions about the molecular mechanism of heme release are unanswered. The authors should therefore tone down their mechanistic speculations. If the authors want to make specific mechanistic statements about the above-mentioned points of the heme binding, transfer and release process, they would have to add supporting experimental data or MD simulations.

RESPONSE: We agree with the reviewer that we should tone down in the discussion on the mechanism of release of heme (holoCcmE). We have revised lines 324 to 341 in the discussion for this, and to better explain the interpretation of data. We cite Fig. 4a, b, c, e, f and Fig. 2d, e that provide Cryo-EM support. We replace mechanism with “hypothesis” when referring to the P-His2 loop. Importantly, we have revised Fig. 6 mechanisms and cycle on CcmABCD to simplify, including discussion of it and the legend. Besides adding ATPase data, we now state that while hydrolysis is necessary, we can not determine which step (hydrolysis, Pi exit) mediates the conformational change of CcmC P-His.

Here is the revised section (page14- 15, line 324-341):

The ABC transporter CcmA₂B₂ has proved not essential for heme transportation to CcmE because the CcmAB genes deletion does not affect holoCcmE formation (ref 24, 25 JMB 2010 and EMBO J 2009). Genetic analyses of more CcmABC variants here confirms that only mutations in CcmC but not in CcmA or CcmB, prevent holoCcmE formation (Fig. S5). We propose that CcmA₂B₂ serves as a mechanical lever to transduce conformational changes from NBDs to CcmB TMDs (transmembrane domains) and further to CcmC via the two coupling helices TM4 and TM5 on CcmC (Fig. 4e). When CcmA₂B₂ is in the closed NBD conformation, the CcmC P-His1 is enabled as an axial ligand to the heme, which, with P-His2 liganding, facilitates the formation of holoCcmE (Fig. 4a, b). When CcmA₂B₂ is in the open NBD state (Fig. 4c), the two CcmBs move away from each other and causes a relative movement between CcmC TM4 and TM5, leading to a significant conformational change in the heme-binding site (Fig. 4f) that breaks the bond between P-His1 and heme, and subsequently allows the holoCcmE to release. An important feature of this release hypothesis involves the flexible P-His2 loop, mediated by CcmC TM5 (Fig. 2d, e). Based on our structures, together with our previous biochemical studies, we propose a mechanism for the ABC transporter release complex CcmABCD (Fig. 6, Supp. video 1).

- The EM density for bound Heme and the sidechain of surrounding residues in the reported structures (closed NBD and semi-open NBD conformations) are not shown (or not shown sufficiently well) in the current version of the manuscript. This is essential information to evaluate the structural analysis.

RESPONSE: We made new figures with cryo-EM density of the key residues. Please refer to Fig. 2a, b.

- Page5, line99-101: The authors report that CcmE was co-expressed with CcmABCD, but not observed in the SDS-PAGE gel. In essence, the authors report an experiment that did not work (complex containing CcmE did not form). It's unclear what the authors conclude from this finding. Also, it's not possible to evaluate the statement because no SDS-PAGE analysis was shown. In general, the biochemical

characterization of the purified protein is somewhat meagre. We suggest that the authors add protein purification and SDS-PAGE gel data to the supplementary figures.

RESPONSE: In vivo it is likely that once holoCcmE is formed, ATP hydrolysis by CcmAB rapidly releases it for the next cycle. Thus nearly all CcmABCD purified from broken *E.coli* cells will be the CcmABCD complex. We have added gel-filtration profile and SDS-PAGE gel data to supplementary figures. Please refer to Fig. S3a.

- Page7, line 136-145, Fig2e and Fig S3: It is reasonable to generate a CcmABCD model with the P-His2 ligand by referring to a homology model. However, any comparisons based on the “predicted model” are rather speculative and would need experimental data to support the mechanistic conclusions drawn (eg, functional data with mutation or another experimental structure with the P-His2 ligand). At present, there is not enough evidence to support the notion that the interactions of TM5 and P-His2 loop of CcmC with CcmB represent the structural basis for holoCcmE release during the ATPase cycle. The mutation experiments tested with Cyt c synthesis function assay and HoloCcmE formation assay can only confirm that H60 and H184 are Heme ligands, but not that the interactions of TM5 and P-His2 loop of CcmC with CcmB represent the structural basis for holoCcmE release during the ATPase cycle. The reported deletion of S187 does not directly support conformational changes.

Response: We agree that “predicted” structures are speculative, which is why our major points on P-His2 as a ligand in CcmC (CcmABCD) are based on homology to the CcsBA family member (Cryo-EM of heme in the site), as shown in Fig. 2, as well as previous spectroscopic studies of P-His1 and P-His2 CcmC variants. These are cited appropriately (see response to rev 2). Cryo-EM structures only represent snapshots of the dynamic changes that occur (e.g. with cycling). We believe that we have done a better job at describing what conclusions are based on cryo-EM data, genetics (e.g. P-His2 defects), or homology (to CcsBA). We also removed “immediately suggests”, changed “propose” to “hypothesize” and indicate “may undergo a conformational change” such as going from snapshot to snapshot.

Revised section (page7, line 138-148):

The CcsBA open state thus provides a framework for revealing the CcmC external binding site structure when both P-His ligands are coordinated to heme (Fig. 2e). Genetic evidence (ref 25 JMB 2010) has indicated that, like CcsBA (ref 27 NCB 2022), CcmC requires both P-His ligands for heme binding and in the case of CcmC for holoCcmE formation. A comparison of CcmABCD structures with heme liganded by P-His1 (Fig. 2d) or both P-His residues (Fig. 2e) to non-heme bound states suggests that the P-His2 loop, as well as CcmC TM5, may undergo a major conformational change. Conformational changes between the various cryo-EM structures of CcmABCD show clear movements in TM5 (see below). We hypothesize that CcmC TM5 and P-His2 loop interactions with CcmB represent the structural basis for holoCcmE release during the ATPase cycle.

- Fig. S4: Synthetase functions of mutants.

There is no expression control shown, for example in the form of a Western blot. Hence, the expression levels of these mutant might differ and it is not clear if the observed reduction in activity is due to decreased expression levels or functional deficiency.

Response: For most of the mutants, heme-staining results showed that the holoCcmE expression level is comparable with that of the WT, thus these mutants have normal expression level of CcmE and

CcmABCD. To test if the inactivity of holoCcmE production is caused by expression level, we expressed the WT and mutant proteins with His-tag on both CcmA and CcmE and did Western blot to test the expression level of the mutant. Our results showed that these mutants had normal expression level of CcmE and CcmA. Please refer to Fig. S3b.

- Page8 177-179: The authors state that in the apoCcmABCD closed conformations, each CcmA binds one inorganic phosphate: The EM density of bound inorganic phosphate is not shown to be evaluated by the reader / reviewer. This information should be provided.

Response: We made figures with EM density for bound inorganic phosphate. Please refer to Fig. S2a.

- Fig. S5: ATP binding sites. Because there are so many labels in the figure, the EM density is difficult to evaluate. Based on the density displayed in Fig. S5, we are not convinced that Mg²⁺ is bound. The authors should make a clearer figure showing the density of Mg²⁺ and the bound nucleotides. Furthermore, in the discussion (page 14 lines 305-306), the author state that semi-open NBDs contain a single bound ATP. The relevant EM density should be clearly shown.

Response: We made new figures to better show the EM density. Please refer to Fig. S2 b, c, d.

Minor comments

- In the Methods section, Page 18 line 419:
The gel filtration buffer containing 20 mM Tris-HCl (pH 7.5) and 50 mM NaCl (no detergent). Could the author confirm this information? Was there really no detergent in this buffer?

Response: We made correction in the text.

- A molecular movie showing the observed conformational changes (e.g. morphing fully open; semi-open and fully closed states) would be helpful.

Response: Excellent idea. We have produced a 30 second video (Supp. video 1) that shows moving from the closed NBD with all substrates to the release of heme (holoCcmE) upon open NBD state.

- Fig 5e: CcmC TM helix labeling is confusing (differences between left and right panel).

Response: We made a new figure (Now Fig. 4e) to make it clearer.

- Page 13, lines 292-298. We feel this text is redundant.

Response: We agree this is redundant with the intro and we now start the discussion with “Step 1 of the System I cyt c maturation pathway is catalyzed by...”

Reviewer #2 (Remarks to the Author):

The article entitled “Structure and Mechanisms of the CcmABCD Heme Release Complex” by Li et al., describes the CryoEM structures of the CcmABCD complex in several different conformations. The complex was investigated with ATP, AMP-PNP, or alone and showed distinct conformations. These along with site directed mutagenesis and protein characterization provides a mechanism for the movement of heme through the CcmABCD complex to CcmE. Overall this provides new information on ATP hydrolysis by the CcmABCD complex for the movement of heme to CcmE. There are a few major concerns. First although the resolution of the CryoEM structure is high (lowest 2.86 Å), there are several claims made throughout the text that would require additional data, such as crystallographic or spectroscopic, to be supported. These include distances measured that are below that of the structural resolution of ligands in the proteins. It is possible that there is existing data to support these observations, but these references are missing from the text.

Response: The resolution of a cryo-EM map is defined by the maximum frequency at which the reconstructed density map is reliable; while the accuracy of the distance between two atoms of the atomic model (PDB file) is a reflection of the ‘center precision’ of the refined atoms, which is always much better than the resolution of the map. During the refinement, the atomic model of a protein structure is refined as a whole under strict stereochemical restrictions including the chemical bonds, secondary structures, dihedral torsion angles of the peptides, side chain distances, hydrogen interactions etc., which can lead to very high accuracy of the centers of individual atoms. In most cases, the RMSD (root mean square deviation) of a fully refined atomic model can be lower than 0.1 Å using a map at 3.0 Å resolution. As we have described in some of the responses to rev, we have provided clearer analyses of structures and densities. We are confident about the distances we present, which are consistent with the resolutions of complexes.

Second, there are multiple formatting and grammatical mistakes. For formatting neither the figures nor the supplemental figures are listed sequentially within the text. Figure 3 is the last figure to be mentioned in the text while Figure 5 is mentioned third in the text. Likewise the supplemental figures proceed from S1 to S5, then S4 and finally S6, S7, and S8. There was no mention of figures S2 or S3 within the text. This makes the flow of the figures and text difficult to follow. Grammatical mistakes are outlined below.

Response: We rearranged the figures and tried our best to cite them in the text in order.

Last, there is lots of data within the paper that is buried and not well explained. Very few of the variants, specifically CcmB, are discussed. An additional sentence or two within specific sections would be useful in further describing the findings. This is particularly apparent for all the variants and makes it hard to understand some of the discussion including lines 306-308. It is unclear how or why deletion of these subunits resulted in a loading of CcmE unless a reference to explain this was omitted. Overall, the manuscript with major revisions to improve the explanation for the data interpretation and how the data support the conclusions drawn.

Response: Thank you for this feedback. We forgot to add references on prior genetic results (in lines 324-326, as described above to rev 1)—we now have added these references. As noted above with reviewer 1 comment, we have better explained the genetics results (line 326-341). Thank you to the

reviewer for pointing this out. We have also modified other sections that hopefully clarify. We have better explained and provided all data requested, as we describe in response to rev 1.

Specific corrections

Page 8, line 179, “full-close” conformation should be “full-closed” conformation. This close to should be changed to closed throughout.

Page 10, lines 228-229, the parenthesis around (of the WWD protein family) should be removed.

Also within the section on page 11 and throughout the manuscript, continue to use the superscript with the subunit letter for all of the variants. This is very useful to the reader.

Page 12, line 278, Beta doesn't need to be capitalized

Page 16, line 357, the word that should be changed to the

Page 16, line 365, Add the word Alternately before CcsBA, to compare the two paths.

RESPONSE: We have made corrections in the text.

REVIEWER COMMENTS

Reviewer #2 (Remarks to the Author):

The revision of the manuscript entitled "Structure and Mechanism of the CcmABCD Heme Release Complex" by Li et al., addressed both the reviewers comments. The authors added substantial data and edited their conclusions to be more consistent with the results. The presentation of the current and previous findings has been improved and the reviewer supports publication of the revised manuscript.

Reviewer #3 (Remarks to the Author):

In the revision, the authors have substantially improved their manuscript. It is also obvious that the proposed mechanism is still largely speculative, due to the missing link between the ATP hydrolysis cycle and heme transfer. Therefore, I would suggest the author NOT to emphasize the mechanism too much. Especially the title could be changed to "Structures of the CcmABCD Heme Release Complex at multiple states" or something similar. I have a few minor comments which might be helpful for the authors to further improve their manuscript for publication.

1. As the authors stated, CcmABCD transfers heme to CcmE in an ATP-dependent manner. Can this reaction be reconstituted in vitro using purified protein? This biochemical assay might be very useful to ascertain that the purified CcmABCD for the structural study is functional in terms of heme transfer, and to dissect the functional coupling between ATP binding/hydrolysis and heme transfer. If there is any technical difficulty to perform a such experiment, the author could cite related literature or state: no such assay has been established.
2. The current mechanistic model is still largely hypothetical. I would suggest changing "elucidate the dynamics behind" at Line 28: to "propose a hypothetical model".
3. Is heme initially synthesized in the cytosol? If this is true, how could heme go from cytosol to the periplasmic space and bind to CcmC? The author might want to explain this in the intro and provide related references.
4. There are so many structures in the manuscript but their appearances are unorganized. The author might want to consider following their hypothetical model in Fig. 6 to restructure the manuscript, by talking about the ATP-bound NBD-dimerized structure first, then the Pi-bound state, and finally the NBD-separated state. In this way, the logic flow might be more straightforward for the readers to understand the proposed mechanistic model.
5. I would suggest using "putative" Pi and Mg for Fig. S2a. The affinity of Pi to ABC transporter is usually low, especially since there is no exogenous Pi in the final SEC buffer or supplemented in the Apo protein for cryo-EM sample preparation.

6. What is the oxidation state of heme? Is it ferrous or ferric? The author might include the UV-Vis spectrum (Soret band) of the protein sample.

7. Please label heme affinity in Fig.6, such as “high affinity” “intermediate affinity” or “low affinity”.

REVIEWER COMMENTS

Reviewer #2 (Remarks to the Author):

The revision of the manuscript entitled "Structure and Mechanism of the CcmABCD Heme Release Complex" by Li et al., addressed both the reviewers comments. The authors added substantial data and edited their conclusions to be more consistent with the results. The presentation of the current and previous findings has been improved and the reviewer supports publication of the revised manuscript.

RESPONSE: We thank the reviewer's positive comments.

Reviewer #3 (Remarks to the Author):

In the revision, the authors have substantially improved their manuscript. It is also obvious that the proposed mechanism is still largely speculative, due to the missing link between the ATP hydrolysis cycle and heme transfer. Therefore, I would suggest the author NOT to emphasize the mechanism too much. Especially the title could be changed to "Structures of the CcmABCD Heme Release Complex at multiple states" or something similar. I have a few minor comments which might be helpful for the authors to further improve their manuscript for publication.

RESPONSE: Thank you to this reviewer for their thoughtful comments. We have changed the title as the reviewer suggested.

1. As the authors stated, CcmABCD transfers heme to CcmE in an ATP-dependent manner. Can this reaction be reconstituted in vitro using purified protein? This biochemical assay might be very useful to ascertain that the purified CcmABCD for the structural study is functional in terms of heme transfer, and to dissect the functional coupling between ATP binding/hydrolysis and heme transfer. If there is any technical difficulty to perform a such experiment, the author could cite related literature or state: no such assay has been established.

RESPONSE: We have not established an in vitro assay for the process: heme binding, oxidation, apoCcmE binding, CcmE covalent adduct formation, then release via ATP hydrolysis.

2. The current mechanistic model is still largely hypothetical. I would suggest changing "elucidate the dynamics behind" at Line 28: to "propose a hypothetical model".

RESPONSE: We have changed the text as the reviewer suggested. Please refer to page 2 line 27-28.

3. Is heme initially synthesized in the cytosol? If this is true, how could heme go from cytosol to the periplasmic space and bind to CcmC? The author might want to explain this in the intro and provide related references.

RESPONSE: Yes, heme is synthesized in the cytosol in bacteria. In the Discussion (lines 383-396) and Fig. 5d we discuss exactly what the reviewer rightly points out is important—"how heme traffics to the external site in CcmC". We feel this is appropriate for the Discussion section and that our analyses is rigorous.

4. There are so many structures in the manuscript but their appearances are unorganized. The author might want to consider following their hypothetical model in Fig. 6 to restructure the manuscript, by talking about the ATP-bound NBD-dimerized structure first, then the Pi-bound state, and finally the NBD-separated state. In this way, the logic flow might be more straightforward for the readers to understand the proposed mechanistic model.

RESPONSE: We thank the reviewer for the suggestion. We described the structures in this logic flow: We start from the overall structure of the CcmABCD and three different conformations, then we focused on the conformation of CcmABCD in the presence of substrates (ATP and heme) and by comparing the conformational changes we proposed the hypothetical mechanistic model.

The reviewer has a point that structures could be presented in a different order, yet the model is based on the structures so our opinion is that either order is appropriate.

5. I would suggest using "putative" Pi and Mg for Fig. S2a. The affinity of Pi to ABC transporter is usually low, especially since there is no exogenous Pi in the

final SEC buffer or supplemented in the Apo protein for cryo-EM sample preparation.

RESPONSE: We have changed the text as the reviewer suggested. Please refer to page 2 line 18-20 of supplementary figure file.

6. What is the oxidation state of heme? Is it ferrous or ferric? The author might include the UV-Vis spectrum (Soret band) of the protein sample.

RESPONSE: The purified protein is heme free and the heme was added exogenously in the presence of DTT when making cryo-EM grids. Previous study (Richard-Fogal et. al, EMBO J. 2009 Aug 19) indicated that the heme iron is likely Fe²⁺ at this stage, yet they did not purify a CcmABCD complex with endogenous heme, and to form the CcmE His130 adduct would need ferric heme. We changed "heme" to "ferrous heme" in page 5 line 96.

7. Please label heme affinity in Fig.6, such as "high affinity" "intermediate affinity" or "low affinity".

RESPONSE: Possibly the reviewer is referring to whether in the CcmC external binding site, heme has two his ligands (likely high affinity but not measured) vs one his ligand, or whether the adduct to CcmE His130 is formed--- in which case heme would be high affinity with respect to CcmE binding. The cartoon diagram displays these ligands and adduct features.